# Polymerization within Nanoporous Anodized Alumina Oxide Templates (AAO): A Critical Survey

**DOI:** 10.3390/polym15030525

**Published:** 2023-01-19

**Authors:** Carmen Mijangos, Jaime Martin

**Affiliations:** 1Instituto de Ciencia y Tecnología de Polímeros, ICTP-CSIC, Juan de la Cierva 3, 28006 Madrid, Spain; 2Donostia International Physics Center, DIPC, Paseo de Manuel Lardizabal 4, 20018 Donostia-San Sebastian, Spain; 3POLYMAT, University of the Basque Country UPV/EHU, Avenida Tolosa 72, 20018 Donostia-San Sebastian, Spain; 4Grupo de Polímeros, Centro de Investigacións Tecnolóxicas (CIT), Universidade da Coruña, 15471 Ferrol, Spain

**Keywords:** nanopolymerization, nanoconfinement, nanoporous geometries, AAO templates reaction kinetics, modeling, polymer nanostructures, polymer properties

## Abstract

In the last few years, the polymerization of monomers within the nanocavities of porous materials has been thoroughly studied and developed, allowing for the synthesis of polymers with tailored morphologies, chemical architectures and functionalities. This is thus a subject of paramount scientific and technological relevance, which, however, has not previously been analyzed from a general perspective. The present overview reports the state of the art on polymerization reactions in spatial confinement within porous materials, focusing on the use of anodized aluminum oxide (AAO) templates. It includes the description of the AAO templates used as nanoreactors. The polymerization reactions are categorized based on the polymerization mechanism. Amongst others, this includes electrochemical polymerization, free radical polymerization, step polymerization and atom transfer radical polymerization (ATRP). For each polymerization mechanism, a further subdivision is made based on the nature of the monomer used. Other aspects of “in situ” polymerization reactions in restricted AAO geometries include: conversion monitoring, kinetic studies, modeling and polymer characterization. In addition to the description of the polymerization process itself, the use of polymer materials derived from polymerization in AAO templates in nanotechnology applications, is also highlighted. Finally, the review is concluded with a general discussion outlining the challenges that remain in the field.

## 1. General Introduction to Polymerization Reactions in Confinement and Scope

The study of the polymerization reactions within spatially confined anodized aluminum oxide templates (AAO) and other nanoscale geometries, i.e., controlled porous glasses (CPG), metal–organic framework (MOF) or mesoporous silica (MS), started almost 30 years ago with the pioneer works of Martin, Bein, Aida and Mallouk, among others [1,2,3,4,5,6,7]. As a result, it has attracted significant interest within the polymer science community because it allows for the easy and high-throughput production of nanostructured polymeric materials, the modeling of specific chemical reactions, e.g., the synthesis of biomacromolecules in biological media, and the assessment of the physical and chemical properties of polymers under spatial confinement [8,9,10,11,12,13,14,15,16,17,18].

Focusing first on the capacity to generate nanostructured materials, the polymerization of synthetic and natural polymers inside nanoscale cavities has enabled the fabrication of 0-D, 1-D, 2-D and 3-D nanostructures [8,19,20,21,22,23,24,25,26,27,28,29,30]. The addition of new physicochemical properties resulting from the large surface-to-volume ratio of these nanostructures to the set of intrinsic properties of polymers has opened a plethora of new possibilities in a number of scientific fields, from electronics to optics, catalysis, biomedicine, energy, etc. Polymer materials and devices that just two decades ago were unimaginable are now being produced, including surfaces with bio-inspired solid adhesion [31], self-operating nanoswimmers [31], photonic [32] or plasmonic antennas [33], whose outstanding properties are derived from their complex nanoscale morphologies. Moreover, the combination of the properties above with the stimuli-responsive properties of polymers offers a route to advanced smart materials, which have the potential to revolutionize the fields of biomedicine, electronics and energy [12,13,34,35,36,37,38,39,40].

In addition to allowing for the production of nanostructured polymers, polymerization reactions within spatially restricted spaces can be used to further understand, and in some cases mimic, the synthesis of biopolymers in biological media. Natural polymers are produced by enzymatic catalysis, where stereoselective, regioselective and chemoselective polymerization reactions occur within well-defined nanoscale “cavities”, which results in tissues with outstanding functionalities [11,12,13,14]. For example, bone tissues exhibit an unusual combination of strength and toughness resulting from a precise selection and organization of materials at the nanoscale: nanometer-sized hydroxyapatite crystals periodically deposit within the gap zones of collagen fibrils, leading to remarkably high elastic modulus along with high energy dissipation and resistance to fracture.

Lastly, gaining an understanding of materials’ properties at the nanoscale is of undeniable academic and practical interest. It is well-known that physicochemical properties of materials—including polymers—change when the material is confined to the nanoscale, the rationale being that the size of the material comes into conflict with certain characteristic length-scales associated with physicochemical processes [22,41,42,43,44,45,46,47,48,49,50,51]. Polymerization in confined geometries therefore allows not only for the investigation of the polymerization reaction itself, but also the production of tailor-made nanoconfined polymers whose physical and chemical properties can be assessed.

Two main experimental strategies have been employed to spatially confine polymerization reactions: the use of soft and hard templates. The soft-template strategy utilizes small-molecules which themselves organize into nanoscale cavities within which the polymerization of the monomers occurs. Examples of this approach are the polymerization in micelles, in lipid bilayers, in liquid crystals, in inclusion complexes, etc. [6,52,53,54,55]. Aida et al., among others, have already addressed polymerizations conducted under soft nanoconfinement, widely demonstrating the feasibility of this approach to fabricate well-defined two- and three-dimensional macromolecular objects, see, e.g., Reference [55] for further details. Our paper will instead report on the state-of-the-art developments on the polymerization within hard AAO templates.

According to most of the reported literature, the reduced space of nanoreactors tends to shorten the reaction time and seems to lead to macromolecules of higher molecular weight and/or lower dispersity [56,57]. Nevertheless, the opposite effect, i.e., a broader molecular weight distribution, has been also reported [58]. In some of these works, mathematical models have been implemented to describe and understand the differences between polymerization in bulk and in confinement in porous nanocavities [59,60]. The study of polymerization reactions in confined spaces has demanded further investigation and has become the focus of more research over the last few years, due not only to the technological interest in such systems but also because of the need to understand and generalize the complexity of reactions caused by confinement phenomena. An idea of the surge in interest in this field is evidenced by the SCOPUS^®^ search “polymerization in confinement” as a function of the year of publication, as shown in Figure 1.

Nevertheless, although there have already been a few reviews published on polymerization reactions conducted in other nanoporous materials (CPG, MS, or MOF), to our knowledge there is no state-of-the-art review that provides a general overview within nanoporous AAO geometries. Therefore, our focus is placed on the use of anodic aluminum oxide (AAO), which is one of the most widely used nanoporous materials in the study of confinement effects on polymerization. The scope of this review on Polymerization reactions within Nanoporous Anodized Alumina Oxide Templates is to provide a general overview of the state-of-the-art of basic aspects of the fabrication of polymeric nanostructures via the “in situ” polymerization of monomers in spatially restricted geometries, using a diverse range of polymerization mechanisms and monomers in particular, an up-to-date account of the progress of different aspects of polymerization in confinement including monitoring of reaction conversion; kinetic studies; modeling and polymer characterization (Mn, Đ, T_g_, etc.). The review starts with a brief description of the particular AAO geometry. The main part of the review is concerned with polymerization reaction in confinement in AAO geometries. Although it is not easy to establish the classification of polymerization reactions, as a first approach, polymerization works summarized the electrochemical polymerization and then polymerization reactions are categorized based on the polymerization mechanisms: free radical polymerization (FRP); step polymerization; ring-opening polymerization; ATRP polymerization and other mechanisms. The studies are further subdivided based on the chemical nature of monomer. The review also highlights a number of potential applications arising from polymerization in AAO geometries addressed to energy, sensing and electronic applications, and surface properties such as wetting and adhesion.

## 2. Anodized Aluminum Oxide (AAO) Templates Geometry

Several kinds of “host” hard geometries have been employed as nanocavities for the synthesis of functional polymers with controlled structures. These include alumina (AAO) templates [9,61] as well as other geometries; microporous zeolites [62]; mesoporous silicates [63,64,65,66]; controlled porous glasses [57]; metal–organic frameworks (MOFs) and porous polymers [67,68,69], or hybrid geometries [56,70,71,72,73]. These systems can provide internal spaces large enough to hold diverse monomers. Nevertheless, there are significant structural differences in the porosity of porous materials, not only in the pore size or pore surface functionality, but also in the interconnection of the pores. Pore geometries that have been reported include tubular, spherical and network-type morphologies, which can be disordered or assembled into ordered arrays. Moreover, the pore size is an important parameter, since smaller-sized pores produce materials with higher surface areas. In addition, the pore surface functionality can be designed by post-synthesis modification processes or through the use of functional monomers [8,74,75]. Therefore, the possibility of controlling the pore structure and functionality of the porous materials has boosted the study of polymerization reactions.

Porous anodic aluminum oxide (AAO) templates have demonstrated tremendous versatility as nanomols to achieve sophisticated polymer nanostructures of various morphologies and sizes from confined polymer melts or solutions obtained by the so-called polymer infiltration process [8,9,13,24,76,77,78,79,80,81,82,83,84,85,86,87]. More recently, AAO templates have also emerged as “precise” nanoreactors where polymerization reactions can be performed, thus extending the possibilities of this nanoprocessing approach to the entire library of polymers, including functional polymers that cannot be dissolved or melted [9].

The fabrication of AAO templates is based on a facile and economic electrochemical anodization of aluminum sheets. AAO templates and AAO membranes can be synthesized in the laboratory (lab-made) but are also commercially available. Currently, there are a number of different companies providing membranes with more or less irregularly ordered pores and a broad pore size distribution. These membranes were primarily designed for filter applications but have also been used in other processes in which the homogeneity of nanocavities is not so crucial.

Figure 2 shows schemes of the nature of a typical AAO template and SEM images of a commercial AAO template and of lab-made templates. See figure caption for details.

Comparing Figure 2B,C, it is easy to observe the difference between lab-made and commercial AAO samples. At the surface of the commercial one, some pores are connected. In addition, the lower section is very different from the upper side because pore homogeneity is not constant across the membrane and a heterogeneous alumina network is formed. For this reason and the experience gained in different groups with instrumental improvements, many laboratories now produce their own lab-made templates. In fact, since the first works of Masuda et al. [76], AAO templates have been the subject of research in a number of laboratories and companies world-wide as a relevant method in the rapid development of polymer nanotechnology providing polymer based nanomaterials with different shapes and morphologies [88]. More recently, commercial alumina templates have improved with respect to the regularity and homogeneity of pores and the range of pore diameters. Readers interested on the topic are directed to a recent review of Jani et al. [8] on the Progress on nanoporous anodic aluminum oxide and to the advances in surface engineering and emerging applications. Subsequent developments have allowed for an increase in the level of complexity of AAO templates and for the preparation of “branched” AAO templates. As has been recently reported, branched AAO templates are the focus of intensive research due to the multi-sized and material-dependent properties they offer. For further information, see the reviews reported by Q. Xu et al. and P. Theato et al., among others [19,21,26].

In the literature, different polymer infiltration methods in AAO templates have been reported for the fabrication of “tailored” polymer-based nanostructures, such as nanorods and nanotubes, as well as more sophisticated hierarchically shaped nanostructures including core-shell, branched, pyramidal or nanostructured surfaces [88]. Nevertheless, in the polymer infiltration process, some drawbacks have been found. For instance, in the preparation of polymethyl methacrylate (PMMA) solid nanofibers or PMMA/Lu_2_O_3_ NPs composite nanorods, long timescales and high temperatures were required for polymer infiltration. On the other hand, the infiltration of polyisoprene (PI) leads to the thermal degradation of the polymer. Therefore, the infiltration process of a polymer melt in the AAO nanocavities is not a suitable route when the polymer infiltration takes a long time (days, in some cases); the polymer degrades as a consequence of the high temperature needed for infiltration or the polymer does not melt, i.e., thermoset polymers.

The in situ polymerization of monomers within AAO templates has emerged as an easy and direct approach to overcome the above limitations and thus to fabricate nanostructures of many polymers that cannot be produced by polymer infiltration methods. Nevertheless, contrary to infiltration methods where the final chemical structure of the obtained polymer is the same as that of the precursor polymer, i.e., stereoregularity, molecular weight, polydispersity, etc., in the situ polymerization method, the final chemical structure of polymerized compound is not easily predictable and needs to be studied case by case, as described in the following section.

## 3. Polymerization Reactions within AAO Templates

The polymerization reaction determines the main characteristics of the resulting polymer molecules, including the chemical structure, the molecular weight and molecular weight distribution, the sequence distribution, etc. As with any chemical reaction, polymerization reactions are strongly dependent on the nature of the reactive species and/or specific reaction conditions, solvents, molecular diffusions/interactions, etc. [89,90,91,92]. Therefore, if a polymerization reaction is conducted in spatially restricted geometries, spatial confinement effects may impact any of the fundamental steps of the polymerization reaction is based on, i.e., initiation, propagation and termination [6,93,94,95,96]. This in turn can result in significant differences in the polymer structure and properties. For instance, in free radical polymerization (half of the industrial production of common polymers), the propagation process is the most relevant step in the growth of chains and, as a consequence, is largely responsible for the final molecular weight of polymers. Therefore, nanoconfinement is expected to substantially impact the propagation step where the diffusion of reactive species is hindered by the nanoscale dimensions of the reactor. Indeed, it has been noted in many free radical systems that strong modifications of the viscosity can be observed in confinement [97,98], along with changes to the nature of the vitrification process [99], the cage effect, and the gel effect, and these changes in the polymerization reactions are coupled with changes to the molecular weight of the chains produced [56,100,101]. In the step-growth polymerization method, bi- or multifunctional monomers react to form first dimers, then trimers, etc., eventually leading to the generation of long-chain polymers, depending on the final conversion. Therefore, any strong reduction of the initial part of the step-growth polymerization or T_g_ depression, as a consequence of the degree of confinement, would modify the reaction kinetics [102].

The first studies on polymerization reactions in confinement were reported by Martin et al. [58] and dealt with the fabrication of conducting polypyrrole nanotubes using AAO templates [4,5,7] and Pallikari-Viras et al. on the polymerization of methyl methacrylate (MMA) in controlled pore glasses (CPG) [57]. Some years later, the polymerization of styrene in porous coordination polymers (PCP) was reported by Kitagawa et al. in 2004 [59] and the ATRP polymerization by Gorman et al. and Takashi et al. in 2008 [60,103] etc. These works only represent a few examples of those reported from the early 1990s until around 2010. Nevertheless, in these papers, neither the influence of confinement on polymerization kinetics nor the rigorous chemical characterization of the obtained polymers was reported. Since the first work on the electrochemically initiated synthesis of divinylbenzene (DVB) and ethylvinylbenzene (EVB) copolymers in porous alumina (AAO) membranes in 1990 [1], a relatively broad range of different polymerization techniques have been reported. These include the radical polymerization of several vinyl monomers, step-growth polymerizations, ATRP and others. With a few exceptions, the polymerization mechanism was assumed to be the same as that of bulk polymerization or, simply, was not taken into consideration.

In this section, we report and discuss the first electrochemical polymerization reaction and the most-relevant polymerization reactions within porous AAO templates classified by the polymerization mechanism. In addition, this section includes some of the previous studies on nanoporous polycarbonate membranes (both sides open) of similar diameter to those of AAO templates.

### 3.1. Electrochemical Polymerization

The first work of electrochemical polymerization reactions in AAO nanocavities was reported by Martin’s group in 1990 [1], although it may be noted that this work was preceded in 1986 by their work on the synthesis and morphological control of electronically conductive polymers in nucleopore polycarbonate membranes [2,58]. Due to the interest of their electronic/physical properties and potential application in electronics devices, this work was later extended to the synthesis of π-electron conjugated polymers. This section briefly summarizes a selection of the work relating to such conjugated polymers including polyaniline (PANI), polypyrrole (PPY) and poly(3,4-ethylenedioxythiophene) (PEDOT) as well as their copolymers. In the majority of these studies, the nature of the polymerization mechanism is not actually referred to. In fact, electropolymerization reactions resemble the electrodeposition reaction of metals, although it is far from being understood and generalized, even taking into account the great amount of works reported in the literature, from the 1st Handbook edition on conjugated polymers: properties, processing, and applications [104], to the last one [105].

In the 1990s, Martin et al. studied the electrochemical polymerization of various ionic monomers under different polymerization and template conditions. In particular, they studied different types of heterocycle monomers, the monomer concentration, the nature of the oxidant and concentration, the polymerization temperature, the reaction time and solvent type. Over the course of this work, they optimized the procedure and adopted a general template-based protocol for the preparation of nanomaterial [106]. The electrochemical synthesis of poly(N-methylpyrrole) nanotubes was first carried out within microporous polycarbonate membranes available in a variety of diameters from 30 to 800 nm, under various polymerization and templates conditions [27,61,107,108]. It was observed that if monomers were polymerized within the nanopores, it was possible to adjust the wall thickness of the resulting polymer by choosing the right polymerization conditions, mainly the polymerization time. Thus, nanotubes with thin walls (at short polymerization times) or thick walls (at long polymerization times) could be obtained, see Figure 3. What they proposed was that for polypyrrole, the nanotubes ultimately “close up” to form solid fibrils. Hence, by controlling the polymerization time, it was possible to make hollow polypyrrole nanotubes or solid fibrils. In addition, by polarized infrared absorption spectroscopy, Martin and co-workers observed that the polymer chains of materials synthesized in the narrowest template were oriented and the conductivity of confined polypyrrole was significantly enhanced. They proposed that the preferential orientation of polymer chains was partially responsible for the observed enhancement in conductivity.

Also related to the synthesis of PPY, some years later, Liu et al. [109] studied the electrochemical polymerization of PPY inside AAO nanopores in order to obtain polypyrrole nanotubes. The authors observed the presence of hollow nanotubes by SEM and suggested that PPY initially deposited on the surface of the pore walls. Moreover, analysis of the membrane surface showed that the nanopores were not completely filled and that they did not stick out of the membrane even after increasing the polymerization time and voltage. The formation of the polymer was confirmed, while the increasing of the conjugation length was observed as a consequence of vibration restrictions caused by the confined environment where the polymer was located.

Building on this work, Esman et al. [110] studied the chemical polymerization of acid-functionalized (COOH) PPY-containing oxidizable monomers using AAO membranes in liquid phase polymerization conditions. They found a compromise between effectiveness and mildness of template dissolution conditions for a safe release of PPY nanotubes, i.e., they found that the effective release of functional poly-COOH–PPY-nanotubes from nanoporous AAO templating critically depended on finely tuned membrane digestion conditions.

Concerning the synthesis of poly(3,4-ethylenedioxythiophene) (PEDOT), Liu et al. [111] reported the fast charging/discharging capability of PEDOT nanotubes by electrochemical polymerization in a porous alumina membrane for high-powered supercapacitor applications (see Section 4.1. for further details). Ji-Woong Back et al. also studied the fabrication and the electrical and optical properties of conducting PEDOT nanotubes by means of vapor deposition polymerization inside AAO templates having a diameter range from 50 to 200 nm [112] (see Section 4.1 for further details).

Related to the templated synthesis of polyaniline microtubules, Pathasarathy and Martin studied the polymerization of aniline in microporous polycarbonate membranes [113]. In contrast to polypyrrole, the polyaniline (PANI) nanotubes did not close up, even after long polymerization times. This was explained on the basis that the surface layers eventually grow across the membrane surface and block the pore ends. As a result, the accessibility of the monomer and the oxidant for entrance into the pores was hindered and further polymerization within the pores was not possible. Therefore, no solid polyaniline fibers could be formed by this method giving rise to an improvement in the conductivity of the confined polymers, compared to the bulk samples, due to the high alignment of the confined polymer chains. Li et al. [114] also reported the synthesis of aniline and pyrrole copolymer nanofibrils within the pores of a microporous AAO membrane. In this case, both polymers were involved in the formation of the copolymer rather than in a composite. Continuing with aniline polymerization, Xiong et al. [115] reported the synthesis of highly ordered polyaniline nanotube arrays by in situ polymerization using AAO as nanoreactors. The doping degree of PANI nanotubes was higher than that of a PANI in bulk, since the crystal form of polyaniline nanotubes resulted in a more ordered structure of the polymer chains.

Blaszczyk-Lezak et al. [116] reported the fabrication of intrinsically conducting PANI nanostructures (ICP) by polymerization of ANI within AAO templates and compared the properties to those of an extrinsically conducting polyvinyldiflurorine (ECP) and multiwall carbon nanotubes (PVDF + MWCNT) obtained by melt infiltration of a PVDF + MWCNT film in AAO nanocavities of the same dimensions. Analysis of the final nanostructures of PANI and PVDF-MWCNT confirmed, in both cases, the formation of well-aligned and uniform rodlike polymer nanostructures, as shown in Figure 4. It was also found that the polymerization rate of ANI was critical and needed to be well-controlled to achieve the desired morphology. Moreover, confocal Raman microscopy demonstrated the formation of the conducting emeraldine salt of PANI through all the length of the AAO nanocavities and broadband dielectric spectroscopy (BDS) the enhancement of the electrical conductivity of polyaniline nanofibers polymerized in the nanocavities of the AAO template. When comparing a nanostructured ECP (PVDF with a 3% of MWCNT) and a nanostructured ICP (PANI), similar conductivity was obtained.

### 3.2. Radical Polymerization

The first attempts at conducting radical polymerizations in AAO nanoreactors were addressed at the synthesis of polymers inside or at the walls of the AAO nanocavities. Nevertheless, the mechanism and kinetics of polymerization process was not studied, neither the chemical nor the physical properties of the obtained polymers were looked at in detail.

Among the first examples of radical polymerization, Kim et al., reported the polymerization of p-phenylene vinylene (PPV) inside alumina geometries of different dimensions (10–200 nm diameter) and in polycarbonate membrane filters [117]. PPV polymers were synthesized inside the nanopores or on the inner surface of templates by the chemical vapor deposition (CVD) polymerization of dichloro-p-xylene monomer and the subsequent thermal dehydrochlorination of the polymer, following the process described by Schafer et al. [118]. After thermal treatment under argon atmosphere, the PPV nanotubes and nanorods obtained were converted to the corresponding carbonized tubes and nanorods. The study of the obtained nanoobjects by SEM microscopy, IR, UV-Vis, Raman and photoluminescence spectroscopies revealed the formation of monodisperse nanotubes and nanorods.

Grim et al. [119] proposed a nondestructive method to replicate the AAO templates by minimizing the adhesion between the nanofibers formed inside the nanopores and the pore walls. First, authors coated the AAO walls with 1H, 1H, 2H, 2H perfluorodecyltrichlorosilane in order to obtain a 40 nm-thick multilayer of the polymerized silane coupling agent. Then photopolymerization reaction of an amine-modified oligoetheracrylate resin was carried out by UV light in the AAO hard templates at room temperature with the aid of a free radical photoinitiator. In order to extract the arrays of synthesized polyacrylate nanofibers from the AAO hard templates, a nondestructive mechanical method was applied. As an example of the surface grafting of functional polymers on AAO template walls, Duran and coworkers [120] studied the surface grafting of thin films of Poly(γ-benzyl-L-glutamate) PBLG within nanoporous anodic alumina (AAO) by surface-initiated polymerization of the N-carboxy anhydride of benzyl-L-glutamate (BLG-NCA) using 3-aminopropyltriethoxysilane (APTES) as a surface initiator. The authors claimed potential applications if a thin film of PBLG chains with multiple functional side chains is surface grafted, since it can offer a nanoporous platform with a very high density of functional sites. In a complementary work, Duran et al. studied the pore diameter dependence of self-assembly and segmental dynamics of poly(γ-benzyl-L-glutamate) peptide nanorods confined in AAO [121,122].

Recently, Lee et al. [123] studied the thermal polymerization of the functional vinyl monomer based on diarylamino groups attached to a rigid and planar fluorene core (VB-FNPD) inside AAO channels with the aim of obtaining nanotubes that could be employed as nanocontainers for Fe_3_O_4_ nanoparticles and thus make organic light-emitting polymer nanotubes. The functional VB-FNPD monomer, which contained diarylamino groups attached to a rigid and planar fluorene core to give the monomer efficient blue fluorescence, was polymerized thermally inside an AAO template. The new poly-VB-FNPD nanotubes were characterized showing their hollow nature and intense blue fluorescence. In addition, the nanotubes exhibited mechanical flexibility and semi-conductivity. The nanotubes were filled with Fe_3_O_4_ nanoparticles to prepare magnetic light-emitting nanotubes, giving new nanocomposites that could be positioned with magnetic forces. The isolation of polymerizable styryl groups from the fluorescence- and HT-active chromophore core is the key feature for the nanotubes to preserve the optical properties of the monomer (Figure 5). The authors suggested that the proposed method allows for the fabrication of nanotubes with desired functions by tailoring the designated monomer in advance.

The above examples briefly summarize, in chronological order, some of the first examples reported in the literature that just imply radical polymerization methods (although were not specified in the works), photopolymerization reaction and other examples that can be considered of interest from a technological point of view.

The following sections describe a detailed study of Polymerization kinetics of acrylic and other vinyl monomers polymerization for different monomers:

Polystyrene synthesis is the first detailed study of the reaction kinetics of free radical vinyl polymerization within AAO nanoreactors [124]. Using azobisisobutyronitrile (AIBN) as an initiator, Giussi et al. studied the polymerization kinetics of styrene at different temperatures inside AAO templates of 35 nm in diameter, and the results were compared to bulk polymerization. SEM studies allowed for the establishment of the morphology of the final polystyrene (PS) nanostructures and confocal Raman microscopy (CRM) to monitor the monomer conversion as a function of time. This methodology was applied for the determination of the polymerization kinetics in confinement in AAO templates and to demonstrate that the reaction rate in the AAO nanoreactor was three times higher than the polymerization in bulk. By SEC characterization, monomodal chromatograms for polystyrene synthesized in the AAO nanoreactors at all reaction times was obtained while a bimodal behavior was observed for polystyrene obtained in bulk. It was also observed that the weight-averaged molecular weights and the polydispersity index of polymer obtained in confinement were lower than those from bulk polymerization. Moreover, compared to the bulk synthesized polymer, ^13^C NMR spectra of PS nanostructures showed differences in the polymer stereoregularity, probably due to differences in the diffusion of growing chains in confinement, that could favor a more stereoregular polymerization.

Polymerization of perfluorodecyl acrylate was carried out by Salsamendi et al. [125] within AAO nanoreactor to study the free radical polymerization kinetics of a perfluorodecyl acrylate monomer (FA) using AIBN as initiator in order to obtain a superhydrophobic nanostructured polymer surface. The kinetics of the reaction were monitored by following the decrease of the C=C stretching signal of the monomer, as shown in Figure 6a. This monomer yields a low Tg polymer that therefore does not exhibit any gel effect during polymerization. A mathematical model was derived to explain the decrease in polymerization rate when the polymerization was carried out in confinement. Authors observed a good agreement between the compartmentalized model and the experimental data. Nevertheless, bulk free radical polymerization proceeded at a much faster rate, Figure 6b. The proposed model accurately predicted that as the diameter pore decreases the probability of termination is increased, as a consequence of the higher local radical concentration and thus resulting in a decrease of the polymerization rate. See Reference [104] for further details of polymerization model.

SEM morphological studies were conducted in order to establish the final nanostructure of the obtained polymer and to demonstrate the ability to replicate the well-defined surface structure of anodic aluminum oxide (AAO) templates by in situ polymerization. Interestingly, the hydrophobic character of Polyperfluorodecyl acrylate (PFA) surfaces as observed by contact angle measurements increased considerably as a function of pore diameter after nanostructuring a fluorinated acrylic polymer using AAO templates. In fact, the contact angle switched from 114° to 159° when increasing the roughness of the polymer surface from planar (bulk) to rough (extracted nanopillars from 250 nm diameter and 1.5 μm length AAO templates). When a slight angle of 8° was applied to the substrate, the water droplet easily rolled off, thus demonstrating a lotus leaf mimetic surface [125].

Polymerization of methyl methacrylate was studied by Sanz et al. [126] in AAO templates as model monomer that shows a gel effect during polymerization. The evolution of the conversion versus time showed the onset of a rapid polymerization and a strong gel effect for all the polymerization temperatures studied leading to a limiting conversion, similar to polymerization in bulk, Figure 7. In addition, it was also observed that the conversion at which the gel effect appears was higher as the reaction temperature increased and that the maximum conversion obtained increased with increasing reaction temperature. Moreover, the initial rate of polymerization was much faster as the degree of confinement increased. Finally, the molecular weight of polymers obtained in the AAO nanoreactors was significantly lower than those obtained in bulk across all the temperatures studied.

In order to explain the differences between reactions conducted under nanoconfinement and in bulk, the authors proposed a mathematical model based on the free volume approach and three different confinement effects. In the initial phase of polymerization, within the AAO template, a faster decomposition of the initiator molecule was assumed, leading to a faster rate of polymerization. That was confirmed by DSC experiments in which the decomposition of AIBN was observed to occur significantly faster than reported values in bulk. Later, the onset of diffusional limitations was observed to occur earlier which was attributed to the higher glass temperature of polymers under nanoconfinement, which was also experimentally confirmed. Finally, due to the lower effective volume experienced by each radical when placed under nanoconfinement in the AAO templates, the pairwise combination of radicals in the diffusion limiting region would occur faster in the nanoconfined system. These confinement effects combined to give a polymer of lower molecular weight and of lower dispersity.

Polymerization of vinyl pyrrolidone. The kinetic study of thermal and photoinduced radical polymerization of vinyl pyrrolidone (VP) in the restricted geometry of AAO templates was reported as an example of the polymerization of a less activated monomer [127]. The thermal polymerization of VP with AIBN as initiator at 60 °C was monitored using NMR. Although the authors observed that the obtained conversion was lower compared to that of the macroscale, the reaction also proceeded up to high conversion, between 77 and 83%, depending on the selected pore diameter. They also observed a pseudo first-order kinetic dependence for the lowest pore diameters studied, thus indicating an increased level of control over the reaction and the elimination of termination and side reactions with decreasing pore diameter. The molecular weight of the polymers, Mn, obtained in AAO templates increased linearly with monomer conversion, between 30,000 g/mol and 111,000 g/mol, and Đ, varied from 1.1 to 1.7. Therefore, the reaction in nanoconfinement led to a rather homogeneous structure of chains while, for bulk polymerization, the authors suggested probable chain transfer to monomer/polymer reactions, thus indicating the polymer heterogeneity and no control over the reaction for the polymerization in bulk. The authors claimed that AAO membranes are an excellent tool to increase control over the polymerization of vinyl pyrrolidone and to produce materials of well-defined properties.

Polymerization of ionic monomers. Recently, the free radical polymerization of a series of ionic monomers in AAO templates was studied by Tarnacka et al. [29,127,128]. First, 1-butyl-3-vinylimidazolium bis-(trifluoromethanesulfonyl) imide ([bvim][NTf2]) was selected as ionic monomer and AIBN as initiator. The progress of the reactions was monitored by DSC and compared to Broadband Dielectric Spectroscopy results. From isothermal DSC measurements at 363 K, the authors detected a pronounced shift of the exothermic peak to shorter times depending on the polymerization process. These results emphasized the acceleration of the reaction as the pore size decreased, as shown in Figure 8. However, for the smallest pore sizes, some saturation effect was visible. The authors also found that the conversion was around 95% for all examined samples, much higher than typical monomer conversion of radical polymerization (usually, in the range of 40−85%), thus indicating a strong influence of the AAO templates on the reactivity of monomeric ionic liquid. Moreover, the molecular weight of the macromolecules synthesized under confinement was higher with respect to polymers synthesized in bulk. Interestingly, they found that the examined nanopolymerization was quite heterogeneous. Consequently, the properties (molecular dynamics and glass transition temperatures) of the materials recovered at the pore walls and in the middle of the pores differed significantly.

In a later work, the same authors [128] studied the influence of small modifications in the chemical structure of the ionic monomer on the properties of synthesized polymers when they substituted the butyl group for an ethyl group and found a dramatic change in the polymer properties. Very recently, the same group studied the influence of a sterically hindered ionic monomer when the polymerization was carried out by free radical polymerization in two kinds of hard confinement systems, AAO nanoporous membranes and high pressure compressed systems, that influence the free volume of polymerization in different ways. The idea was that the two systems would act as additional driving forces to polymerize sterically hindered monomers in relatively short times. The authors selected imidazolium-based ionic monomer 1-octyl-3-vinylimidazolium bis(trifluoromethanesulfonyl) imide ([OVIM][NTf2]) [129] as the sterically hindered ionic monomer and found noticeable differences between nanomembranes and high-pressure systems, although, for both systems, a significant increase in the control of the reaction and the polyelectrolyte properties was found. Moreover, they also found that the molecular weight increased when the reaction was carried out in AAO nanocavities of 35 and 150 nm in diameter, although the dispersity was slightly broader (Đ: 2.2–2.7) compared to bulk polymerization.

Copolymerization of butyl methacrylate and 2-hydroxyethyl acrylate. Leon et al. extended the study of polymerization reactions in confinement within restricted geometries of AAO templates to the free radical copolymerization of butyl methacrylate (BMA) and 2-hydroxyethyl acrylate (HEA) monomers in order to obtain BMA-HEA copolymer nanostructures with tunable mechanical characteristics and swelling/wetting properties [130,131]. For an initial molar composition of 0.45/0.55, with AIBN as the initiator, at 70 °C, the copolymerization reaction within AAO templates of 70 nm diameter led to high conversion (100%) (Figure 9). Moreover, the BMA-HEA copolymer obtained under confinement showed a similar molecular composition to the initial one. The reactivity ratios of the copolymerization reaction were determined for the P(BMA/HEA) copolymer synthesized in confinement at different molar compositions. HEA is the most reactive monomer in the copolymerization reaction in confinement and in bulk, however, in the restricted geometry of the AAO template, the reactivity ratios shift slightly away from the preferential incorporation of HEA. In addition, Young’s modulus and wetting behaviors were significantly modified, and the obtained BMA-HEA copolymer nanopillar structures showed interesting soft surface and significant swelling capacity.

### 3.3. Step-Growth Polymerization

Choi et al. [132] studied the fabrication of high-aspect-ratio uniformly bent polymeric nanopillars of polyurethanes that could be used as a physical adhesive. To do so, they carried out the reaction of a functionalized prepolymer with an acrylate group acting as crosslinker on surface-treated reusable AAO molds. They also studied the collapse and sticking properties of polymer nanopillars as a function of the dimensions of the nanocavity. Related to stepwise polymerization in confinement, Mavaldi et al. reported a theoretical study on the effects of confinement in diffusion-controlled stepwise polymerization by Monte Carlo simulation [133]. They found that confinement modified both the spatial pair distribution function and the diffusive properties of the polymers and, therefore, that the system can experience either faster or slower reaction kinetics when compared to bulk reactions, depending on the strength of intermolecular interactions. This model has been used in experimental works of step polymerization in AAO templates, as will be described later.

The first kinetic study of the step-growth polymerization reactions in AAO membranes was carried out by Tarnacka et al. and Maksim et al. using epoxy based systems [134,135]. The authors studied the step-growth polymerization of bisphenol-A diglycidyl ether (DGEBA) with aniline both in anodic aluminum oxide (AAO) membranes and in bulk. The authors found [102] that polymerization was faster under confinement compared to the analogous reaction carried out in the bulk system at the same temperature conditions. They also found that the reaction accelerated with the degree of confinement and that the initial step of the polymerization was significantly reduced or even suppressed in nanochannels, i.e., the kinetic curves did not follow the sigmoidal shape characteristic for this kind of autocatalytic chemical reaction.

Very recently, Sanz et al. studied step-growth polymerization within AAO templates [136] and concomitant confinement effects on the reaction kinetics of triethylene glycol (TEG) and hexamethylenediisocyanate (HDI) for three different pore diameters: 140, 60 and 35 nm. The authors found that the kinetics of the reaction in AAO nanoreactors were faster than in bulk, while the molecular weight and the dispersity of extracted polymer from AAO template were reduced. In that work, a mathematical model was developed taking into account the effects of confinement and chemical interactions of OH with the pore walls (Figure 10). Briefly, it was assumed that diisocianate also reacted with the OH of the AAO walls, and the polyaddition in nanoconfinement was catalyzed by hydroxyl groups on the pore wall, resulting in an increase in the rate of polymerization during the early stages of the reaction. The model explained the differences found for both the kinetics of step-growth polymerization and the molecular weights of the obtained polymers in confinement in comparison to bulk.

### 3.4. ATRP Polymerization

Atomic transfer radical polymerization (ATRP) has been also used for the synthesis of different polymers and copolymers. The possibility of combining AAO templates with controlled/”living” radical polymerization techniques leads to more controlled polymerization systems. In the following section, some representative examples are reported chronologically.

ATRP copolymerization of NIPAM-co-MBAA. Yue Cui et al. studied, for the first time, the synthesis of thermosensitive Poly(*N*-isopropylacrylamide)/*N*,*N*′-methylenebisacrylamide (PNIPAM-co-MBAA) nanotubes by atom transfer radical polymerization within AAO porous membranes [137]. The ATRP polymerization of NIPAM-co-MBAA copolymer was surface-initiated on aminosilane surface-modified porous anodic aluminum oxide membrane as shown schematically in Figure 11. The copolymerization was successful and led to polymer nanotubes with high thermosensitive behavior. Moreover, in an aqueous environment, AFM studies in real time revealed that changes in the temperature induced changes in the dimension and shape of the nanotubes, i.e., upon heating to LCST or above, the nanotubes underwent a shape alteration from elliptical to circular in water. The same authors also studied the synthesis of PNIPAM-*co*-MBAA copolymers with various copolymer compositions [138] and the influence of monomer concentration on the size of nanotubes. Through SEM, TEM, AFM, and GPC measurements, the authors showed that an increase in the monomer concentration in the polymerization system was followed by a proportional increment of thickness of the nanotubes.

In the following sections, the ATRP polymerization and copolymerization of a series of monomers are also reported [60,138,139,140,141,142]

ATRP Polymerization of MMA. Gorman et al. studied the polymerization of methyl methacrylate by surface-initiated polymerization [60] and the effect of substrate geometry on the polymer molecular weight and dispersity when chains were grown on anodically etched aluminum oxide (AAO) substrates via surface-initiated atom transfer radical polymerization. The cleavage of chains from the substrates using hydrogen fluoride was evidenced by the evolution of Si-OH, C-H, C=O and Si-O bands from infrared spectroscopy. The authors found that PMMA grown on AAO substrates had a much lower molecular weight and a broader molecular weight distribution than polymer grown in solution, and they proposed that confinement effects imposed by the pores during the polymerization would reduce growth rates and increase the polydispersity of chains. Moreover, Kelly et al. studied [143] a modified sol−gel technique to continuously vary the pore diameters in porous alumina templates by means of ATRP grafting of PMMA, and Wang et al. studied the surface-initiated ATRP polymerization of acrylic acid on dopamine-functionalized AAO membranes [141].

ATRP Polymerization of NIPAM. Li et al. [140] prepared a series of thermo-responsive “gating” membranes with controllable length and density by grafting poly(*N*-isopropylacrylamide) (PNIPAM) chains onto the pores of AAO membranes by atom-transfer radical polymerization (ATRP) mechanisms. First, they immobilized 3-aminopropyl)triethoxysilane (APTES) on the AAO nanopores. This was followed by post-modification with the initiator 2-bromoisobutyryl bromide (BIBB) and the subsequent polymerization of NIPAM from the surface. The effect of grafting temperature and time, the feed concentration of NIPAM and the density of –Br on the grafting yields of AAO-*g*-PNIPAM membranes was studied in order to control the thermo-responsive efficiency of AAO-*g*-PNIPAM gating membranes. The study was carried out by checking the diffusional permeation of vitamin B12 at temperatures above and below the lower critical solution temperature.

ATRP Copolymerization of NIPAm-AAm. Sanz et al. [74] studied the synthesis of poly(N-isopropylacrylamide-acrylamide) (PNIPAm-AAm) microgels of different chemical compositions by surface initiated-atom transfer radical polymerization (SI-ATRP) on anodized aluminum oxide (AAO) templates with pore diameter and length of 200 nm and 700 nm, respectively. Previously, AAO walls were silanized, and AAm was used as a co-monomer in order to tune not only the lower critical solution temperature (LCST) but also the surface mechanical properties and the hydrophilicity of the copolymer, as shown in Figure 12. The hypothesis of a possible local molecular rearrangement of PNIPAm-AAm nanostructures at the LCST was proposed. Below the LCST, the copolymer nanopillars would present a random distribution of monomers in the polymer network, but, when the temperature increased, the aggregation of NIPAm segments would occur near the center of the nanopillars, displacing hydrophilic AAm units towards the surface. Molecular simulation results supported the hypothesis that as temperature increased above the LCST a higher concentration of hydrophilic units near the surface would lead to enhanced hydrophilicity.

In a later work, Giussi et al. [35] studied the polymerization of NIPAm-AAm in the presence of Fe_3_O_4_ nanoparticles within AAO templates with the objective of obtaining thermo-responsive PNIPAm nanopillars displaying enhanced responsiveness through the incorporation of nanoparticles. The final objective was to obtain a material that combines more than one stimulus-responsive property into a single material. When increasing the temperature above the volume phase transition temperature, the incorporation of magnetic nanoparticles into the PNIPAm-AAm nanopillars sharply increased the stiffness and hydrophobicity of the samples, the magnetic response being proportional to the amount of nanoparticles incorporated.

### 3.5. Other Polymerization Reactions

In the literature, only a few examples of coordination and ring-opening polymerizations within AAO templates have been reported.

#### 3.5.1. Coordination Polymerization

It is worth mentioning the work of Choi et al. on the synthesis of syndiotactic polystyrene (sPS) [144] using a metallocene catalyst within AAO membranes in conjunction with methylaluminoxane cocatalyst carried out in a silica modified AAO template with the aim of obtaining nanofibrils of ultrahigh molecular weight. First, the pore surfaces of these AAO films were coated with silica by the surface sol-gel method, afterwards the metallocene catalyst was supported onto the inner walls of a silica nanotube reactor to make a silica-supported metallocene catalyst (anchoring the catalyst onto the surfaces of porous silica particles). For the polymerization of styrene, the catalyst-deposited silica AAO membrane was placed over a liquid mixture of styrene and *n*-heptane of different concentrations and the reaction carried out at 70 °C. The authors found very thin sPS nanofibrils (<10 micron) grown at the catalytic sites on the pore walls that aggregated to form intertwined, rope-like nanofibrils with diameters in the range of 30–50 nm, which further intertwined into even larger 200 nm diameter polymer nanofibrils. The sPS synthesized in the silica nanotube reactor (SNTR) has an ultrahigh molecular weight (Mw: 928,000 g/mol) with a large fraction of 2,000,000–5,000,000 g/mol molecular weight polymers [144].

The synthesis of polyethylene was carried out by heterogeneous Ziegler-Natta polymerization within nanochannels of anodized aluminum oxide (AAO) membranes [75]. The catalyst was chemisorbed at the inner wall of the nanocavities and monomers diffused from the outside. Results indicated the formation of a highly stressed crystalline structure attributed to catalytic production of excess amounts of polyethylene inside the nanoconfined templates.

#### 3.5.2. Ring-Opening Polymerization of Caprolactone

The ring-opening polymerization of ε-caprolactone inside AAO membranes of 35, 100 and 150 nm has recently been reported and the results compared to those of polymerizations conducted in bulk [145]. For all of the samples studied, independent of pore diameter and examined system, an exothermic peak of polymerization, being 100-times shorter when compared to the macroscale conditions, was observed. The obtained polycaprolactone (PCL) showed an unimodal peak with Mn up to 53.5 kg/mol and a moderate dispersity (Đ = 1.27−1.41), suggesting that side reactions were successfully suppressed due to the applied confinement. In this case, hydroxyle groups from AAO walls would act as both catalyst and initiator of the ring-opening polymerization. Finally, for the polymerization of ε-caprolactone in AAO templates, a pseudo-living coordination−insertion mechanism was proposed.

#### 3.5.3. Other Non-Classified Polymerizations

In addition to those systems already described, during the last two decades some papers dealing with alternative polymerization systems in AAO geometries have been reported that cannot be easily defined. Most of them refer to applications [127,146,147,148]. For instance, Al-kayisi et al. [146] reported the synthesis of crystalline nanorods of 9-anthracenecarboxylic acid, methylene ester (9AC-ME) by photopolymerization of adjacent monomers to form the polymer poly(9AC-ME) deposited over a commercially available alumina membrane of 200 nm of pore diameter and 60 µm of thickness. Irradiation of the devices at wavelengths below 400 nm led to crystal-to-crystal photopolymerization. Although not demonstrated in the study, the authors proposed that crystalline polymer nanorods of different diameters could be fabricated using alumina templates of variable pore diameter.

## 4. Properties and Applications of Synthesized Polymers with AAO Templates

The properties of nanostructured polymeric materials result from the physicochemical features of the nanoconfined polymer (e.g., chemical functionalities, solid-state microstructure, molecular dynamics, etc.) but also from their nanoscale topography. As a result, polymerized nanowire arrays often exhibit different properties compared to their bulk counterparts, which opens up a plethora of new applications for these materials. In the following section, we review some of the most recent research developments in the application of polymerized nanowires for energy storage, sensing and electronics, as well as functional surfaces and cell adhesion.

### 4.1. Energy, Sensing and Electronic Applications

Supercapacitors are a class of energy-storage device that exhibit high power density (as compared to regular batteries) and large energy density (as compared to regular capacitors). Supercapacitors strongly benefit from the increase of the interface area between the electrolyte and the electrode, as this enhances the charge storage capability and the charge–discharge rates. Therefore, materials systems with large specific surface areas, such as nanowires arrays (once released from the AAO template), have attracted significant attention for electrode construction in supercapacitors.

Polymers which are π-conjugated are the preferred polymeric choice for the preparation of electrodes in supercapacitors owing to their different redox states. For example, Cao et al. grew polyaniline nanowires with various morphologies by changing the concentration of supporting electrolyte and the monomer, the potential of electropolymerization, and the polymerization time [149]. They observed that the increase of the sulfuric acid concentration during growth induced a transition from solid nanowires with tubular nanostructures, also finding a clear correlation between the electrochemical capacitance of the nanostructures and their morphology. Strikingly, they obtained a specific capacitance value of 700 F/g (measured at a charge/discharge rate of 5 A/g) for nanowires grown at 0.5 M aniline concentration. Liu et al. [111] fabricated a symmetric supercapacitor employing electrochemically grown poly(3,4-ethylenedioxythiophene) (PEDOT) nanotube electrodes and probed its energy density and power density. The device reached a remarkable power density of 25 kW kg^−1^ while maintaining 80% energy density (5.6 W h kg^−1^), which was rationalized assuming that counter-ions can penetrate into nanotubes and thus access internal polymer surfaces. Duay and coworkers also synthesized a flexible pseudocapacitor, this time asymmetric, comprising polymerized PEDOT nanowires as anode and a PEDOT/MnO_2_ nanowire array as cathode [150] (Figure 13). The cell exhibited a total capacitance of 0.26 F and a maximum voltage of 1.7 V and is thus capable of powering small portable electronics.

Polymer nanowire arrays also find application as chemical sensors, because, like supercapacitors, they also benefit from a large surface area, as this increases the material’s interaction with the probed analyte. Nanowires of π-conjugated polymers offer the possibility to sense chemical species (e.g., H^+^ and gases) due to their good electrochemical reversibility, good chemical stability and electrical conductivity that can change through interactions with a number of molecules [151,152,153]. For example, Sulka et al. synthesized hydroquinone monosulfonate-doped polypyrrole (PPy-HQS) nanowires within AAO templates by potentiostatic electropolymerization of a monomer in a solution containing 0.05 M pyrrole + 0.05 M potassium hydroquinone monosulfonate and a basic electrolyte [154]. They tested the efficiency of the nanowire arrays as potentiometric pH sensors and, interestingly, the PPy-HQS nanowires exhibited better performance than on PPy-HQS thin films. A similar nanowire sensor was achieved by Xhang et al. by electropolymerizing pyrrole monomers within AAO templates [155]. The device exhibited a high response to the detection of low ammonia concentrations and a comparatively short response and recovery time. Dan et al. also measured the sensing response of PEDOT/PSS nanowires to various vapors, including acetone, methanol and ethanol [156]. The conductivity of the nanowires was 11.5 ± 0.7 S/cm, and the contact resistance was 27.6 ± 4 kΩ. In the presence of acetone, methanol and ethanol (at the saturated vapor pressures), the wires showed a resistance change of 10.5%, 9%, and 4%, respectively.

Since the pioneering works by Martin [108,157,158], a number of studies, including the electrical characterization of π-conjugated polymer nanowires, have been reported. In general, an increase of the electrical conductivity has been observed for nanowires, resulting from a preferential alignment of polymer chains or crystals within the nanowires. Moreover, it seems that the larger the spatial confinement imposed, the larger their conductivity. For example, Shirai et al. found the electrical conductivity of electrochemically synthesized PEDOT nanowires had strong size dependency, with the conductivity increasing as the nanowire diameter decreased [159]. From the estimated doping levels (∼5%) and conductivity data (∼100 S/cm), they claimed that the charge carrier mobility of the nanowire with the smallest diameter amounted to 2.0 cm^2^/V s, which is four orders of magnitude larger than that of the bulk PEDOT material (4.0 × 10^−4^ cm^2^/Vs). Along the same lines, highly conductive (PEDOT) nanotubes, exhibiting conductivity values of 2000 S/cm (for a single nanotube as measured by conductive scanning probe microscopy) were synthesized by vapor deposition polymerization by Back et al. [112]. For that, they first deposited an oxidant layer on the pores by wetting the pore walls with 10% *w/w* solution of FeCl_3_·6H_2_O in THF. Then, the functionalized AAO templates were exposed to an EDOT monomer vapor for about 10 min in a vapor deposition chamber under ambient conditions. Polyaniline nanowires with diameters ranging from 20 to 100 nm were chemically synthesized by Wang et al. within AAO membranes [160]. They analyzed their electrical properties as well as their field emission properties, for which they observed a low threshold electric field (5–6 V/μm) and maximum emission current density of ∼5 mA/cm^2^, which suggest the applicability of this material for the development of field emission displays. Lee et al. produced semiconducting nanotubes by thermal polymerization of the monomer conjugated VB-FNPD ([123]). They characterized the I-V response of single nanotubes and calculated conductivity values of 3.2–5.6 × 10^−6^ S cm^−1^. Lastly, Tarnacka et al. used BDS to probe charge transport of the polymerized ionic liquid 1-butyl-3-vinylimidazolium bis-(trifluoromethanesulfonyl)imide poly([bvim][NTf2]) confined within AAO nanopores [29]. They first studied how the polymerization process proceeded in the confined medium and interestingly, they also observed that nanoconfined polymeric ionic liquid had a higher electrical conductivity than same material obtained by bulk polymerization.

### 4.2. Surface Properties: Wetting and Adhesion

Superhydrophobicity is an important property for the development of self-cleaning, anti-corrosive or anti-fogging surfaces. This can be readily achieved by inducing multiple-scale roughness in a low surface energy material [161]. Polymer nanopillars and nanotubes, such as those resulting from templating AAO matrices, offer the ideal topography for the development of superhydrophobic surfaces as they mimic the ultimate level of roughness of some naturally occurring superhydrophobic surfaces, such as that of lotus leaves. This has triggered the study of wetting properties of these kinds of nanostructures in the last years. For example, Zhang et al. produced lotus-leaf-like surfaces with two different level of roughness by polymerizing styrene end-functionalized perfluoropolyether (PFPE) and a highly fluorinated styrene sulfonate ester (SS) within the nanopores of a pre-patterned AAO template. They conducted a thorough analysis of the wetting properties of these surfaces and demonstrated that they exhibited superhydrophobic characteristics, low contact angle hysteresis and self-cleaning features. In addition, they also performed theoretical calculations that suggested that these properties can be rationalized by assuming that water droplets are in the Cassie state on the nanopillar arrays. Nanostructured surfaces with two roughness levels were also produced by Bok et al. employing an elegant approach [162]. First, they electrochemically polymerized PPY inside the nanopores. After the nanowires were released, Au nanoparticles were synthesized onto them by chemical methods. Finally, in order to decrease the surface energy of the material, the decorated nanowires were coated with heptadecafluoro-1-decanethiol. The system thus had two roughness levels that could be independently adjusted: a submicron-scale roughness resulting from the nanopillars and a nanoscale roughness from the Au nanoparticles. This allowed them to effectively tune the wettability of the surface. Qu et al. electropolymerized polyaniline within nanoporous AAO films that were supported on Ti/Si substrates. After removal of the template, the polyaniline nanowires remained attached to the Ti/Si substrate, which enabled the fabrication of superhydrophobic and electrically conductive surfaces [163]. Bessonov et al. presented the fabrication of acrylate-based surfaces with controlled wettability that included alternative superhydrophobic and hydrophilic regions [164]. Hydrophobic and hydrophilic regions were achieved by UV curing the resin employing a segmented master that contained pieces of AAO templates. Salsamendi et al. polymerized a intrinsically low surface energy fluorinated acrylic monomer (MFA) in AAO nanopores and measured a remarkable contact angle of 159° for the released nanostructured surface [125] (see Section 3.2 for further details).

Nano- and microscale topography is known to be a key factor governing successful scaffold–cell interactions as cells are able to sense nano- and microscale features through specific responses [165,166]. Thus, nanowires/nanotube arrays seem to be ideal platforms for investigating the interactions between cells and nano/microscale environments with the final goal of achieving topographical cellular control [167]. Recently, Lastra et al. prepared nanowires of β-Ketonitrile tautomeric copolymers using this approach, and after investigating their biocompatibility, cytotoxicity and degradation properties, they found that nanostructured scaffolds exhibited a higher degradation rate than non-nanostructured counterparts, with the value being compatible with typical bone regeneration times [168]. Furthermore, the β-Ketonitrile tautomeric copolymer nanowire arrays also permitted osteoblastic cell proliferation and differentiation without cytotoxic effects (Figure 14). Hong et al. prepared EpCAM antibody-coated polypyrrole nanowires doped with NHS-SS-biotin with the aim of producing a surface with the ability to capture, quantify and release circulating tumor cells [169]. To do so, they first electrochemically polymerized pyrrole monomer containing the NHS-SS-biotin within the pores. They proved that nanowires functionalized with biotinylated anti-EpCAM effectively captured cells, and this platform could be effectively used to electrochemically identify and detect cancer cells employing horseradish peroxidase (HRP)-labeled and anti-EpCAM-conjugated nanoparticles (HRP/anti-EpCAM Ppy NPs).

In addition to the above, other applications have been also explored for polymer nanowires and nanotubes synthesized inside the nanopores of AAO templates. For example, Grimm et al. managed to produce nanopillar arrays of UV photopolymerized polyacrylates with high morphological quality that were proposed for use as nanomolds in order to pattern nanoporous surfaces in a second material, e.g., silica (122). The polyacrylate nanopillars could be mechanically detached both form the original AAO and the nanoporous silica, paving the way for the high-throughput fabrication of nanostructured surfaces.

## 5. General Discussion and Conclusions

Significant effort has been devoted to developing in situ polymerization reactions in AAO templates and in other nanoporous hard materials of different geometries (non-reported in the present review). The vast majority of reported works have addressed either the influence of spatial confinement on the polymerization process, in particular, on reaction kinetics and molecular weight of the synthesized polymer or have sought to obtain polymer nanostructures with modulated properties. Free radical, step-growth and ATRP polymerizations are the most studied polymerization reactions. Although some of the reported results are contradictory and do not seem to follow a general trend, even for the same template or reaction mechanism, some general conclusions can be drawn:

Constrained AAO geometries as well as mesoporous silica templates, porous controlled pore glasses and metal–organic framework can be considered as nanoreactors for the polymerization reactions in order to obtain polymer nanostructures directly from the precursor monomer. Polymerization reactions in nanopores tend to yield a high degree of conversion with higher rates than the comparative reaction in bulk polymerization. In many cases, it reaches near 100% conversion. Experiments of polymerization in confinement show that each system is different, and depends on the mechanism of polymerization process—radical polymerization, polycondensation, or other mechanism—and on the porous material—AAO templates, mesoporous silica, controlled pore glass nanopores, etc. The T_g_ of the polymers synthesized in confinement are in general higher than their bulk counterparts. Nevertheless, the molecular weight does not follow a general rule, and both higher and lower values have been reported compared to the bulk.

Although the present review only refers to AAO templates, the comparison with other confining media is not trivial, as the degree and the geometry of confinement exerted in by each system is different. For example, while AAO templates can be considered isolated nanoreactors, the nanopores in CGP are interconnected. For this reason, general conclusions for nanopolymerization reactions are difficult to drawn.

Anodized aluminum oxide (AAO) templates present some advantages for studies of polymerization reaction. For instance, the kinetics of polymerization reaction in confinement can be monitored in situ. Confinement impacts free radical polymerization, yielding lower molecular weights and dispersity values. In some cases, polymerization within the AAO template favors the polymer stereoregularity. Modelling of polymerization has been successfully implemented for radical polymerization, including for those monomers that show a pronounced gel-effect, and also for step-growth polymerizations. In both cases confinement effects have been incorporated into the models in order to describe the experimental data.

Indeed, polymer nanostructures can be readily fabricated from monomers in a faster and less-energetic process than those obtained by polymer infiltration process. In particular, (i) the long polymer infiltration time (hours or days) can be overcome by the in situ polymer synthesis in 1–2 h; (ii) the high temperatures required for polymer infiltration (near 200 °C) and concomitant polymer degradation effect can be avoided since polymerization temperature is around 50–80 °C in most cases; (iii) thermoset polymers that cannot be infiltrated in AAO templates monomers (polymer does not melt) can be synthesized directly from the monomers in the AAO templates (most of monomers are liquid).

Therefore, polymerization within AAO nanopores can be considered a complementary method to that of the infiltration of pre-formed polymers (nanomoulding) to obtain tailored polymer nanostructures. In fact, from polymerization methods, (i) polymer nanorods/nanofibers nanostructures with different morphologies and nano-microscopic sizes (pore diameter ranging from 35 to 350 nm and pore length ranging from 700 nm to 100 µm) can be synthesized directly inside the AAO nanoreactors and then extracted from the nanocavities; (ii) the entire library of polymers can be synthesized from the precursor monomers by polymerization reactions inside AAO nanocavities. This includes thermoset polymers. Moreover, dual-sized polymer nanowires mimicking the design of gecko toe pad can be obtained from branched AAO templates.

AAO nanoreactors have been employed as precursor of applications. The construction of electrochemical biosensors that show good operational stability, polymer scaffolds for cell adhesion or “patterned” surfaces, are only a few examples of intricate polymer nanostructures that have been obtained by AAO templating, microlithography or a combination of both.

The aforementioned results support the idea that nanoporous materials are ideal systems to accomplish polymerization reactions in nanoconfinement. However, they also highlight the necessity of further studies, especially on the influence of particular geometrical constraints in each type of polymerization reaction and for different monomer families. Further effort should be also devoted to developing theoretical models that describe these polymerization reactions. Although these fundamental studies are still lacking, the huge progress that has been made in recent years gives promise for the targeting of the new applications for these materials, particularly in systems that can take advantage of the well-defined multilevel polymer architecture that can be achieved via the polymerization of monomers in nanopores.

## Figures and Tables

**Figure 1 polymers-15-00525-f001:**
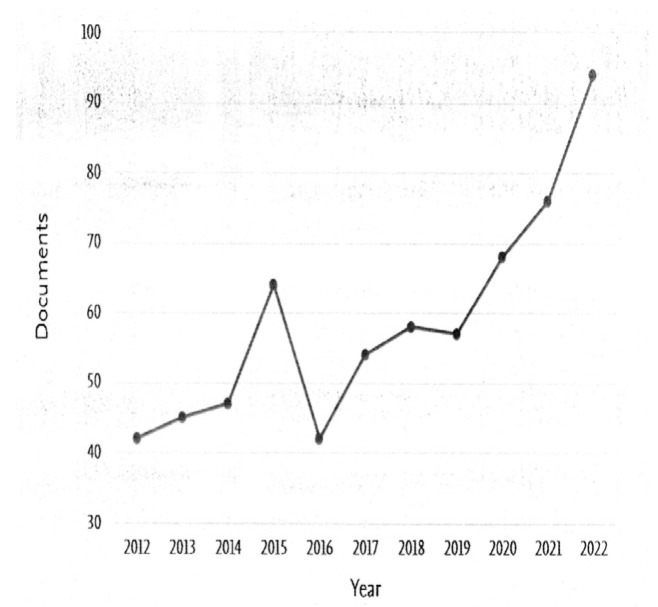
Scopus search for “polymerization in confinement”.

**Figure 2 polymers-15-00525-f002:**
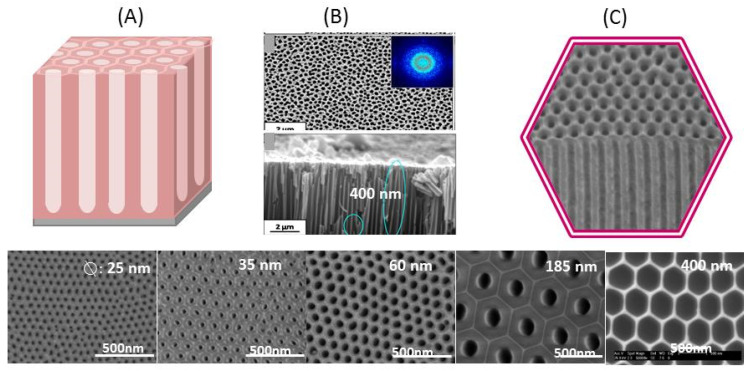
Top panels: Scheme of AAO templates (**A**); SEM images of a commercial (**B**) and a lab-made AAO template (**C**). Bottom panels: SEM images of top views of lab-made AAO templates with different pore diameters, from left to right: 25, 35, 60, 185 and 400 nm.

**Figure 3 polymers-15-00525-f003:**
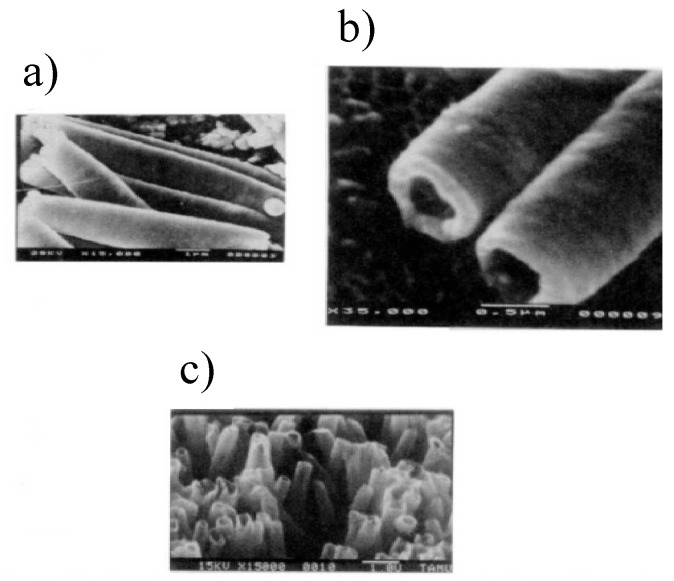
Scanning electron micrographs of chemically synthesized polypyrrole microtubules (**a**,**b**) and electrochemically synthesized poly(N-methylpyrrole) microtubules (**c**) [27], Copyright 2011. Reproduced with permission from the publisher: Springer.

**Figure 4 polymers-15-00525-f004:**
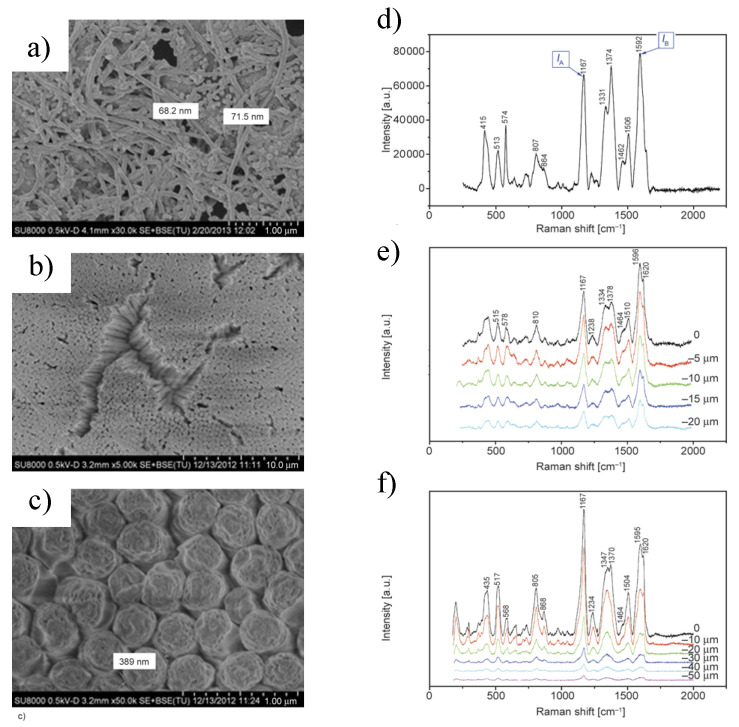
(**a**) Different magnifications of PANI into 45 nm AAO pores (**a**) and 300 nm AAO pore (**b**) and (**c**); (**b**) Raman spectrum of PANI bulk (**d**), PANI polymerized in AAO templates of 300 nm (**e**) and idem in 45 nm (**f**) [116]. Copyright 2016. Reproduced with permission from the publisher: Budapest University of Technology and Economics.

**Figure 5 polymers-15-00525-f005:**
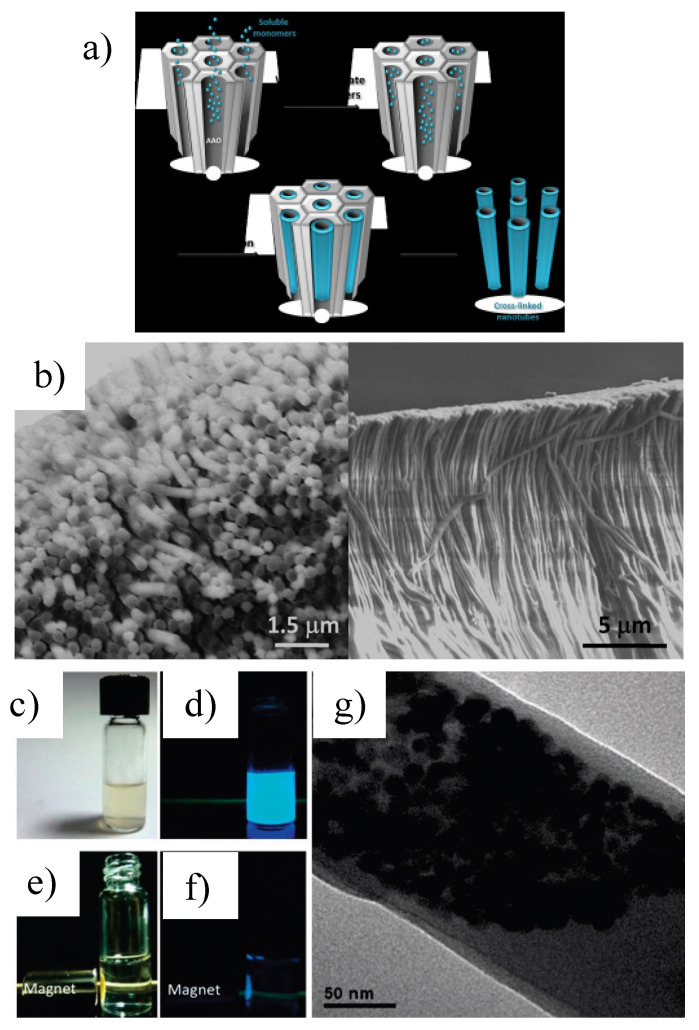
(**a**) Scheme of preparation of poly-VB-FNPD nanotubes in AAO-templated in situ polymerization. (**b**) SEM images of the array of poly-VB-FNPD nanotubes. (**c**) The Fe_3_O_4_NP–OLET nanocomposites dispersed in DI water. (**d**) The corresponding blue emission excited using a 365 nm UV lamp. (**e**) The nanocomposites attracted to one side of the vial by a magnet and (**f**) excited by UV. (**g**) The TEM image of Fe_3_O_4_NP–OLET nanocomposites [123]. Copyright 2014. Reproduced with permission from the publisher: RSC.

**Figure 6 polymers-15-00525-f006:**
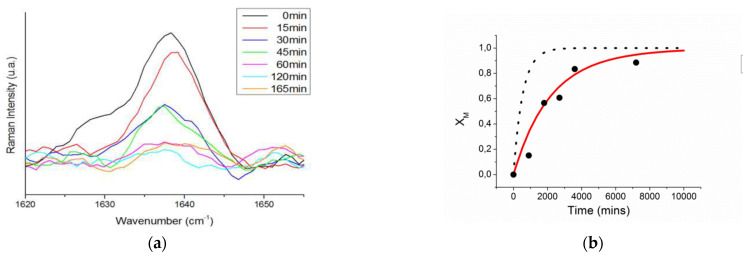
(**a**). The Raman spectra of in situ polymerized PFA in AAO templates as a function of time. (**b**) Graph of conversion versus time for polymerization of PFA in AAO template. Symbols represent experimental points while the dotted and solid lines represent the model prediction for bulk and nanoconfined polymerization, respectively [125]. Copyright 2015. Reproduced with permission from the publisher: RSC.

**Figure 7 polymers-15-00525-f007:**
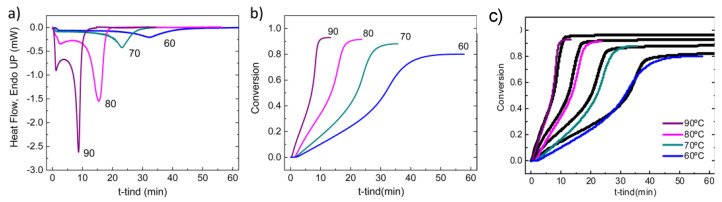
(**a**) Heat flow measured in the nanoconfined polymerization of MMA in AAO 60 at different temperatures under isothermal conditions. (**b**) Evolution of conversion with time. (**c**) Kinetic plot of MMA polymerization conducted in AAO templates with nanopore of 60 nm diameter at 60, 70, 80 and 90 °C. The bold lines indicate the fit of the kinetics to the mathematical model described herein [126]. Copyright 2017. Reproduced with permission from the publisher: ACS.

**Figure 8 polymers-15-00525-f008:**
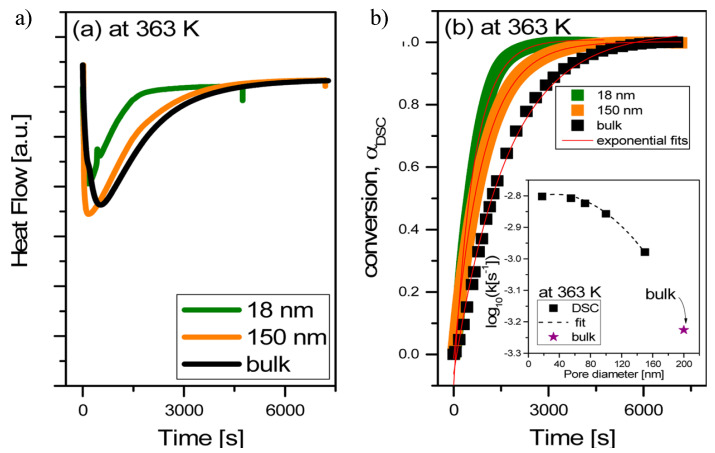
(**a**) Raw isothermal DSC data. (**b**) Time evolution of the monomer conversion determined from DSC measurements. As an inset in panel (**b**), the dependence of the polymerization rate constant vs. pore diameter is presented [128]. Copyright 2016. Reproduced with permission from the publisher: RSC.

**Figure 9 polymers-15-00525-f009:**
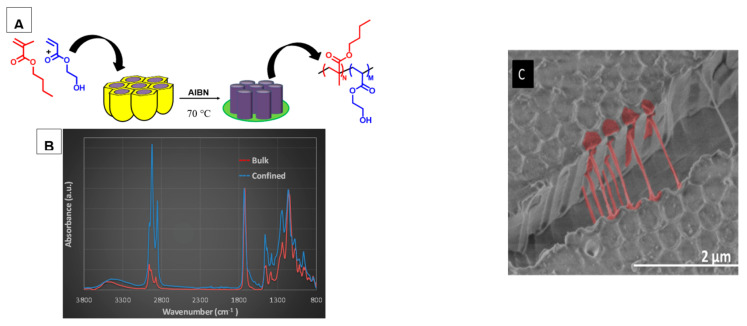
(**A**) Schematic copolymerization procedure of BMA and HEA monomers under confinement. (**B**) Infrared spectroscopy (IR) spectra of the copolymers obtained under confinement and in bulk. (**C**) SEM image of BMA-HEA copolymer nanostructures [130]. Copyright 2019. Reproduced with permission from the publisher: MDPI.

**Figure 10 polymers-15-00525-f010:**
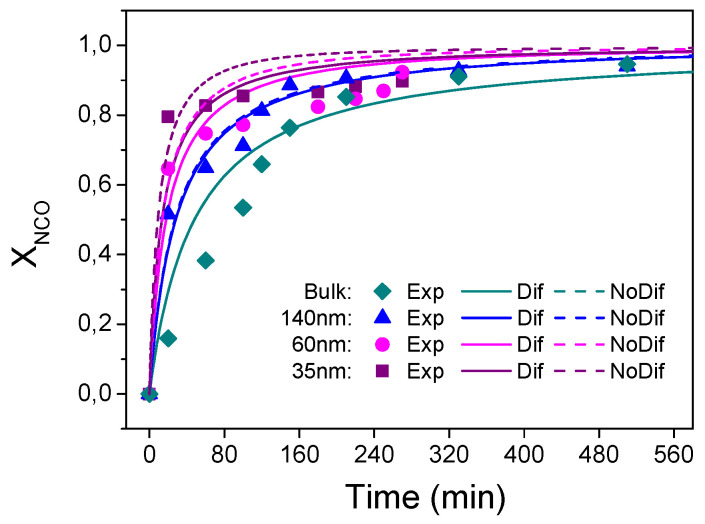
Conversion versus time of polyurethanes polymerized in bulk, as well as in 35, 60 and 140 nm pores at 80 °C. Symbols indicate experimental data, and solid and dotted lines show diffusion (Dif) and no diffusion (NoDif) model predictions ([136]). Copyright 2018. Reproduced with permission from the publisher: Elsevier.

**Figure 11 polymers-15-00525-f011:**
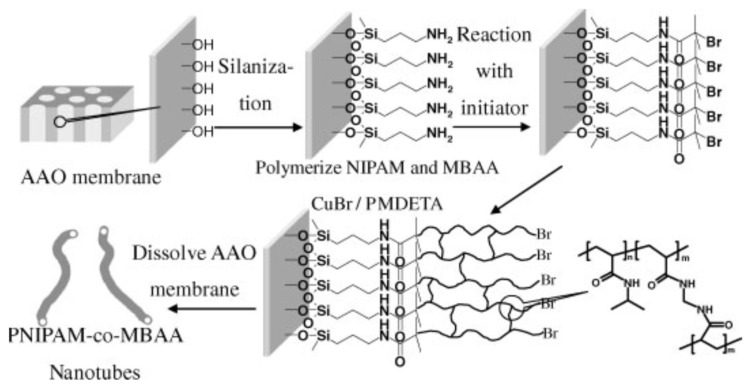
Schematic illustration of the fabrication of PNIPAM nanotubes in a porous AAO membrane [137]. Copyright 2005. Reproduced with permission from the publisher: Wiley.

**Figure 12 polymers-15-00525-f012:**
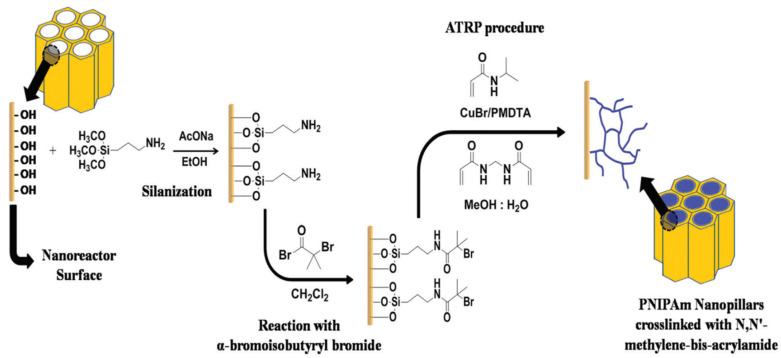
Scheme of synthesis of PNIPAm-AAm nanopillars [74]. Copyright 2017. Reproduced with permission from the publisher: RSC.

**Figure 13 polymers-15-00525-f013:**
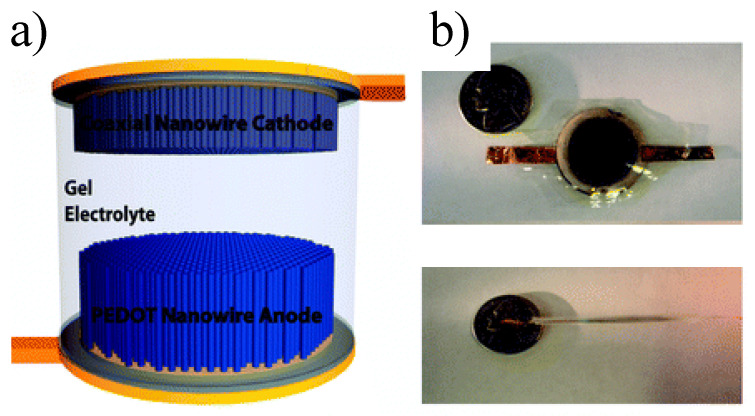
(**a**) Side-view schematic of the asymmetric device, where gray represents the coaxial MnO_2_/PEDOT nanowire cathode and blue represents the PEDOT nanowire anode (**b**) Picture of the same demo device [150]. Copyright 2012. Reproduced with permission from the publisher: RSC.

**Figure 14 polymers-15-00525-f014:**
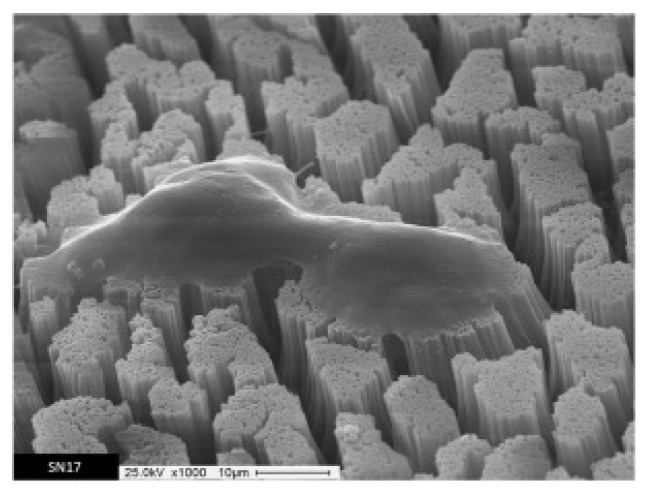
SEM image of an osteoblastic cell on β-Ketonitrile tautomeric copolymer nanowires [168], Copyright 2015. Reproduced with permission from the publisher: Elsevier.

## Data Availability

The data used to support the findings of this study are available from the literature and the corresponding author upon request.

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
