# Peer review of "Polymerization within Nanoporous Anodized Alumina Oxide Templates (AAO): A Critical Survey"

_polymers, 2023, doi:10.3390/polym15030525_

Round 1

Reviewer 1 Report

The review presented by the authors is devoted to the study of the processes and mechanisms of polymerization of monomers in nanopores of matrices based on aluminum oxide, as well as their characterization. The authors have done a lot of analytical work aimed at studying this issue, as well as considering various types of templates based on alumina, including various polymerization mechanisms. In general, this direction is very promising, which is also noted by the authors, and quite a lot of work has been devoted to the problem of filling the pores of templates for creating new types of materials. The authors in their review tried to summarize all currently known works in this area and structured them in one work. This review can be used for further research as one of the fundamental works, but before accepting it for publication, the authors should answer a number of questions.

1. First, it is necessary to give a brief justification of the reasons for choosing aluminum oxide as template matrices in most cases, and today a fairly large number of other types of templates, including polymer membranes, are known.

2. The above analysis of publication activity according to the Scopus database should be redone, the authors should use several keywords to search for scientific publications, then the list will be significantly expanded. You should also add a mention of the use of template matrices based on aluminum oxide.

3. Authors should provide more details about exactly what structures can be obtained using these matrices and methods.

Reviewer 2 Report

The manuscript presents a review of the synthesis of polymers with the use of anodic alumina oxide (AAO) templates to grow polymers in a confined way. The objective is to use the AAO confined growth in order to change the polymer morphologies, densities, functionalizations and chemical architectures. A detailed review is made, starting from the AAO templates formation and controlled geometries (pore sizes, interpore distances and so on). Then, the polymerization reactions inside the AAO pores are reviewed, with a detailed discussion of the synthesis of several polymers according to the reviewed polymerization method (electrochemical, radical, step-growth, ATRP and other polymerization methods). Some of the properties and applications of the developed AAO synthesized polymers is reviewed. The manuscript has an original review on the subject and only needs minor revisions. I have the following comments:

- The manuscript relies on many acronyms (for example: of polymer names) as is normal in this area. However, acronyms should be defined before they are used. The manuscript is inconsistent in this respect. Some acronyms are never defined (e.g., VB-FNPD or NIPAM, are used, but never defined) and some are used, but only defined latter (e.g., PNIPAM is used first in page 16 but is only defined in page 17). This makes the text harder to read. The authors should carefully revise the manuscript in order to identify the meaning of all the used acronyms, for clarity to the reader.

- On page 11 when the authors write that “the reaction in the AAO nanoreactor was three times higher” are they referring to the reaction rate ?

- On page 21, when the authors write “with the conductivity increasing as the nanowire decreased” are they referring to the nanowire diameter decrease ?

- Figure 1 has small and somewhat blurred images. It should be improved. The a) figure in figure 6 is missing. Is it the one on the right of figure 6 ? The caption of figure 9 has a strange character overlapping it, so that it is unreadable. Figure 10 has “Dif” and “NoDif” inside it. What are they ?
